# Oro-Dental Manifestations in a Pediatric Patient Affected by Helsmoortel-Van der Aa Syndrome

**DOI:** 10.3390/ijerph18178957

**Published:** 2021-08-25

**Authors:** Massimo Petruzzi, Alessandro Stella, Valeria Capra, Maria Contaldo, Fedora della Vella

**Affiliations:** 1Interdisciplinary Department of Medicine, University of Bari “Aldo Moro”, 70124 Bari, Italy; 2Department of Human Oncology and Biomedical Sciences, University of Bari “Aldo Moro”, 70124 Bari, Italy; alessandro.stella@uniba.it; 3IRCSS Istituto G. Gaslini, 16147 Genoa, Italy; valeriacapra@gaslini.org; 4Multidisciplinary Department of Medical-Surgical and Dental Specialties, University of Campania Luigi Vanvitelli, 80138 Naples, Italy; maria.contaldo@unicampania.it

**Keywords:** Helsmoortel-Van der Aa syndrome, autism, molar incisor hypomineralization, dental abnormalities

## Abstract

*Aim*: Aim of this case report is to describe oro-facial abnormalities in a patient affected by Helsmoortel-Van der Aa syndrome, a rare autism syndrome, with not well described dental and cranial malformations. *Case Report*: Helsmoortel-Van der Aa Syndrome is a rare autosomal genetic syndrome causing mental impairment and autism, craniofacial dysmorphism, chest deformity and multiple organs dysfunction. Oro-facial involvement in Helsmoortel-Van der Aa syndrome has not been thoroughly described yet. The present article reports a case of a 9 years old male patient affected by Helsmoortel-Van der Aa Syndrome, presenting with oral breathing typical facies, high arched palate, II class and dental crowding. The patient teething was adequate to his age. The enamel of incisors and molars showed demineralization areas and dark spots, a clinical picture consistent with molar incisor hypomineralization syndrome. These hypo-mineralized areas are more susceptible to cavities, in fact the patient’s 4.6 tooth was decayed. The child was brought to our attention due to a mucocele on the lower lip, confirmed by histopathologic examination. Available data on oro-dental manifestation of this syndrome are rather poor and inconsistent, also due to the rarity of the disease. The finding of enamel abnormalities in the presented case could suggest a potential genetic etiopathogenesis linked to the same genes causing Helsmoortel-Van der Aa syndrome.

## 1. Introduction

Helsmoortel-Van der Aa syndrome (HVDAS), also known as mental retardation autosomal dominant 28 (MRD28) or ADNP-related intellectual disability and autism spectrum disorder (ADNP-related ID/ASD), is an autosomal dominant monogenic syndrome caused by de novo variants in the last exon of activity-dependent neuroprotector homebox gene (*ADNP* gene), a human transcription factor, essential for brain development. The prevalence of ADNP syndrome is approximately (1–2)/100,000 individuals [1]. The syndrome represents 0.17% of the autism spectrum disorder cases. Helsmoortel et al. described for the first time in 2014 the syndrome, detailing the clinical characteristics, intellectual disability and facial dysmorphisms of 10 patients with HVDAS [1]. According to the ADNP kids research foundation database, there are 205 HVDAS children diagnosed to date [2,3]. The syndrome is characterized by a clinical set of neurobehavioral anomalies, including intellectual disability and autism spectrum disorder. Renal anomalies, hand and feet abnormalities, nail abnormalities, widely spaced nipples, pectus excavatum, pectus carinatum, or combined excavatum/carinatum deformity, cardiac defects, short stature, hormonal deficiencies, gastrointestinal, visual, and musculoskeletal problems are also reported with variable percentages of prevalence.

Craniofacial dysmorphisms give similar facial features in HVDAS patients. They include a prominent forehead, high hairline, eversion or notch of the eyelid, broad nasal bridge, thin upper lip, and smooth/long philtrum. Ear malformations (small or dysplastic, low-set, and posteriorly rotated ears) were observed in nearly half of individuals. Diagnostic pathway and therapeutical management must include speech therapists, physiotherapist, neuropsychiatric evaluation of sleep and behavioral disorders, ophthalmologic, auditory, cardiac, and hormonal surveillance [4].

Few data have been reported on oro-dental manifestations in HVDAS patients. We present a case of pediatric HVDAS patient highlighting the syndromic oro-dental features comparing them with the ones previously reported in the literature, in order to spread the awareness of this condition, consider a possible link between genetic mutations and dental abnormalities and promote oral health in these patients.

## 2. Case Report

When the patient came to our attention, he had recently received a diagnosis of HVDAS. The patient’s parents referred his clinical history and presented the child’s medical records.

Born as a first child, male, with natural delivery after 40 weeks of gestation weighting 3.580 kg and with a length of 52.5 cm. The mother was healthy while the father presented with β-thalassemia and allergy to β-lactam antibiotics. The parents referred the absence of complications during the pregnancy.

At 6 months the child was diagnosed with thalassemic microcytosis and reduced growth, due to agalactia. At 2 years old (y.o.) he was hospitalized due to metabolic acidosis and hypoglycaemia with vomit, and on that occasion, doctors hypnotized a diagnosis of psychomotor delay. The neuropsychiatric evaluation confirmed the presence of hyperkinesia, agitated state and low attention. The brain MRI highlighted hypophysis hypoplasia and presence of an arachnoid cyst in correspondence of *cisterna ambiens*.

The cardiologic and gastroenterological evaluations did not report alterations. The otolaryngologist reported mild adenoid and tonsillar hypertrophy, with oral breathing. Later, the child was diagnosed with encephalopathy, microcephaly, craniofacial dysmorphism, behavioural alterations and cognitive impairment. Risperidone was prescribed at a dosage of 1.25 mg per day.

At 3.5 y.o. he underwent a comparative genomic hybridization array test (Array CG, Human Genome CGH Microarray kit 180 K; Agilent Technologies, Santa Clara, CA, USA) and a multigene panel for cortical dysplasia to detect eventual genetic abnormalities. Both resulted negative.

Next, a whole exome sequencing (WES) genetic test was performed when the child was 7 y.o., using a SureSelect Clinical Research Exome V2 (Agilent Technologies, Santa Clara, CA, USA) on a library prepared using the SureSelect Target Enrichment System (Agilent Technologies, Santa Clara, CA, USA) following the manufacturer’s protocol.

Insertions/deletions (indels) and single nucleotide polymorphisms (SNPs) were called using the Genome Analysis Toolkit (GATK). Using the GATK Variant Quality Score Recalibration approach, indel and SNP calls were filtered and annotated using Variant Effect Predictor. Variants with a minor allele frequency (MAF) > 0.01 based on 1000 Genomes Project (www.internationalgenome.org (accessed on 15 July 2021)), dbSNP 126, 129 and 131 (www.ncbi.nlm.nih.gov/sns/ (accessed on 15 July 2021)), ExAC (http://exac.broadinstitute.org/ (accessed on 15 July 2021)), gnomAD (https://gnomad.broadinstitute.org/ (accessed on 15 July 2021)), were excluded from the analysis. Finally, AlamutVisual^@^ (Sophia Genetics, Boston, MA, USA) was used to annotate and assess the pathogenicity of candidate variants.

WES analysis disclosed the presence of a de novo variant, c.158G > A, p. (Trp53Ter), in exon 4 of the *ADNP* gene.

The variant, present in heterozygosis, has not been reported previously, and was confirmed by means of Sanger capillary sequencing on an ABI 310 automatic sequencer (Applied Biosystems, Waltham, MA, USA). Therefore, the result of WES analysis supported the clinical diagnosis of a neurological disorder compatible with Helsmoortel-Van der Aa syndrome. In fact, the nonsense mutation identified is similar to the nine de novo truncating mutations identified by Helsmoortel et al. [1]. At the age of 8.3 y.o. the parents referred regression of the child’s behaviour, with frequent vomit episodes and daytime sleepiness. The neurological exam highlighted psychomotor instability and anxiety state, a severe cognitive impairment and speech-comprehension deficiency. Walking difficulties was encountered too.

The risperidone dosage was increased to 2 mg per day and theanine supplement was added to therapy.

The patient came to our attention at the Dental Unit of Policlinico of Bari when he was 9 years old for the presence of a mucocele on the lower lip (Figure 1). The intraoral exam highlighted the presence of high-arched palate, dental second-class and dental crowding (Figure 2, Figure 3 and Figure 4). The dental exchange appeared consistent with the patient’s age and sex. Permanent incisors and molars and deciduous second molars presented with discoloured and hypo-mineralized areas, suggesting a clinical diagnosis of molar-incisor hypomineralization (MIH), an enamel qualitative developmental defect with a poorly defined aetiology [5] (Figure 5, Figure 6 and Figure 7). A cavity was also found on the 4.6 tooth. The Decayed Missing Filled Teeth (DMFT) score was 1.

The patient showed complete lack of cooperation, therefore an orthopantomography x-ray could not be performed. The patient was treated under general anaesthesia. The tooth with cavity was treated and filled with composite resin, and the lesion was removed for biopsy with a 15c scalpel.

The histopathological exam confirmed the clinical diagnosis of mucocele. No complication arose after the surgery.

## 3. Discussion

Oral aspects in HVDAS patients are widely described, including teeth abnormalities, soft tissue alterations and jaws dysmorphisms. Although our patient did not show a thin upper lip, this aspect was described in the original report of Helsmoortel et al. in four of ten patients [1]. The high narrow palate, also reported in one patient (#8) of Helsmoortel original paper, the Class II and the labial incompetence reflect the mouth breathing, due to turbinate hypertrophy. Van Dijck et al. reported that one third of HVDAS have widely spaced teeth while in our case we noted a dental crowding due to loss of mesiodistal space [6]. An important aspect of HVDAS is the dentition timing. A correlation between HVDAS and early deciduous teething has been postulated. A recent study found that 81% of children with ADNP syndrome have early deciduous teeth eruption, never described in any other known syndrome [6]. These findings may help to identify children at risk in their first year of age, a time that is extremely early in autism diagnosis. The *ADNP* gene also regulates the teething through the BMP gene family [7].

MIH’s aetiology has not been totally explained yet. Genetic and environmental factors may both play a role in causing this condition. The hypo-mineralized areas of enamel in MIH present an overload of proteins which bind hydroxyapatite crystals interfering their correct organization [8].

Alterations of the teeth enamel are reported in multiple congenital syndrome which include autism spectrum disorder, Timothy-like syndrome, congenital chronic intestinal pseudo-obstruction, and West syndrome [9,10,11].

Probably, also the MIH observed in our patient is genetically related to ADNP mutation. Genes such as enamelin (*ENAM*), tuftelin interacting protein 11 (*TFIP11*) and tuftelin 1 (*TUFT1*) are associated with the development of MIH. Although underreported, MIH could be a further dental sign of *ADNP* gene pathway alteration. In fact, Pascolini G. et al. reported unspecified “dental anomalies” with multiple caries [12]. Enamel anomalies could make the tooth more susceptible to dental caries, also considering the mental impairment and the difficulty of following correct oral hygiene procedures in these patients [13]. In this specific case, MIH were noted on both primary and secondary dentition, while generally it is most observed only in one of them [5].

Oral movement problems, with implications for feeding and speech, were common (45.6%) and were significantly more common in individuals with mutations in the nuclear localization signal and in the C terminal of this domain [12].

Genetic conditions like HVDAS predispose to a number of oral issues, like dysgnathias, tooth abnormalities and cavities. Most common oro-facial alterations reported in patients affected by HVDAS are reported in Table 1. Due to the few available data, there are not oro-dental manifestations shared by all previous case reports, but abnormalities about the space-condition and the occlusal development seems common, while enamel and dental alterations were described only three times (by Helsmoortel et al. [1], Pascolini et al. [12] and the present study).

Behavioural and cognitive retard are also found in these patients, making difficult to detect and to treat oral and dental pathologies as outpatients. Correct information and cooperation between doctors and parents are important to maintain an optimal oral health status for these patients and to prevent more serious diseases [13]. Finally, the mutation identified in the subject described by Van Dijck [6], is the closest to the amino-terminal portion of the ADNP gene so far reported and the only lying in the exon 4 of the gene.

In the other reported cases of HVDAS no genetic mutation falling in exons 1–4 was found, and also oro-dental alterations were lacking, in contrast to the present case report. The clinical features present in our patient could suggest a contributing role of mutations in the ADNP gene exon 4 in the etiopathogenesis of dental abnormalities.

## 4. Conclusions

Patients affected by ADNP syndrome like HVDAS always represent a clinical challenge, first to get a proper diagnosis and later to receive proper care in any medical field. The oro-dental management of these patients is often made difficult due to their lack of cooperation, a key issue of a majority of dental procedures. A higher awareness of these conditions as well as better knowledge of their clinical features, can improve the oral health status and avoid more invasive treatments, improving the quality of life of these patients. Moreover, the hypothesized link between MIH and ADNP syndromes, due to a shared genetic involvement, could help to better understand how premature neuroectodermal mutations influence the phenotype of ADNP syndromes on multiple levels.

## Figures and Tables

**Figure 1 ijerph-18-08957-f001:**
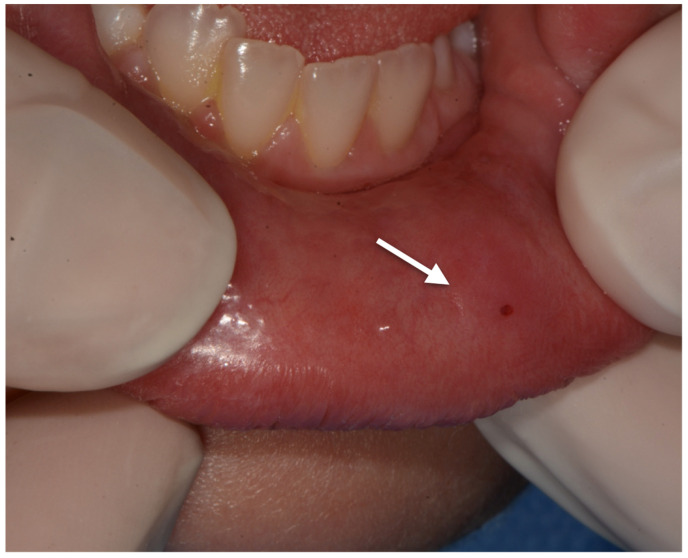
Lower lip mucocele with superficial ecchymosis due to chronic trauma.

**Figure 2 ijerph-18-08957-f002:**
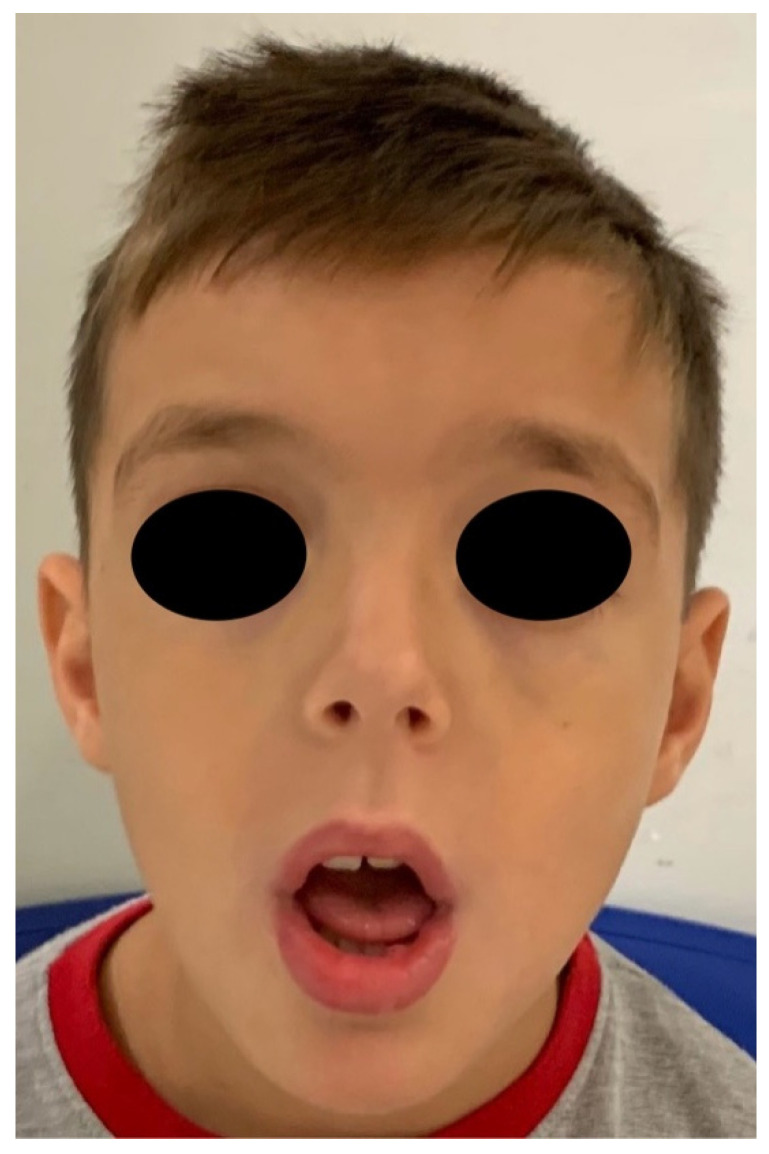
Helsmoortel-Van der Aa Syndrome patient’s facies.

**Figure 3 ijerph-18-08957-f003:**
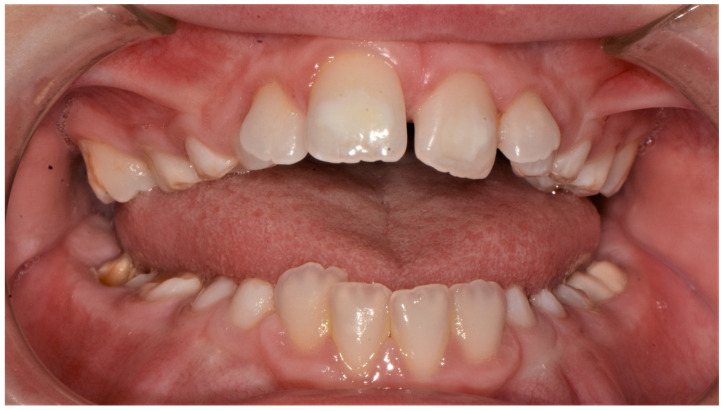
Intraoral picture.

**Figure 4 ijerph-18-08957-f004:**
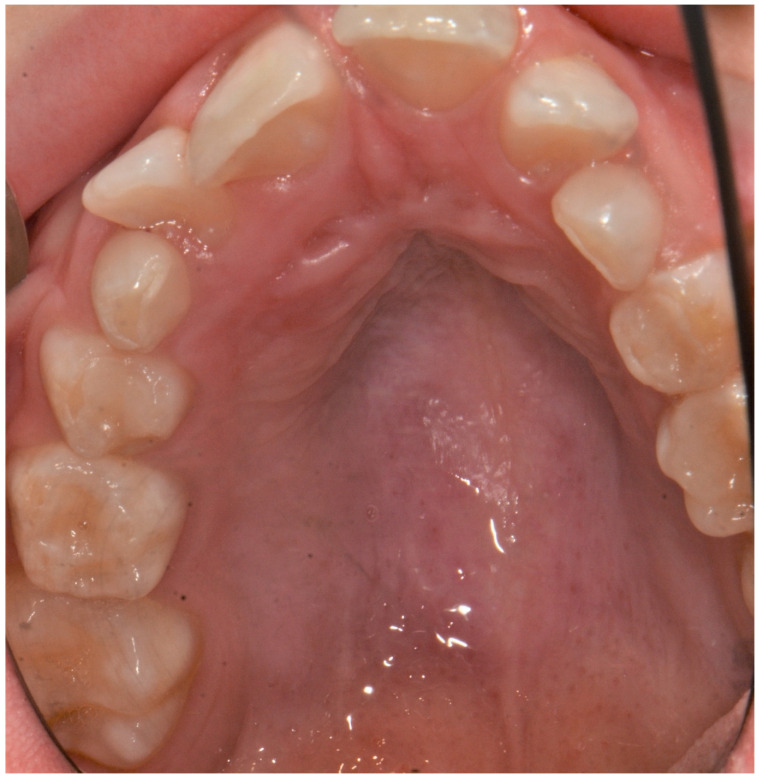
High-arched palate.

**Figure 5 ijerph-18-08957-f005:**
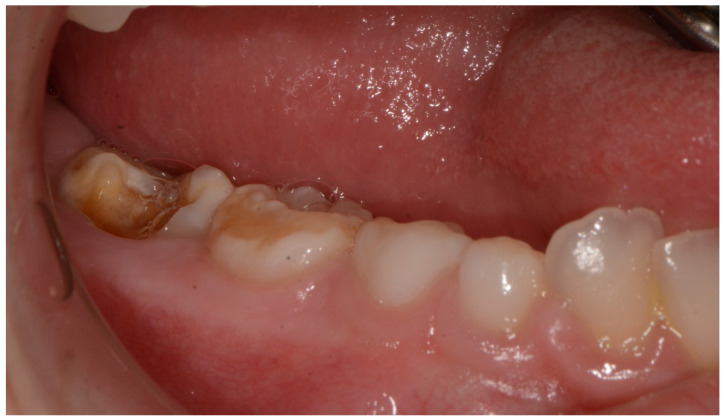
Molar Incisor Hypomineralization details on 8.5 and 4.5 teeth and cavities on 4.6 element.

**Figure 6 ijerph-18-08957-f006:**
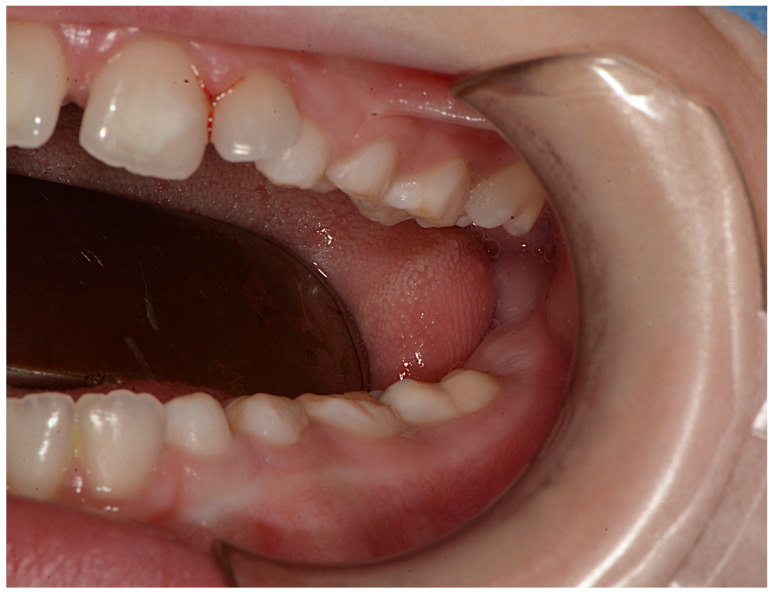
Molar incisor hypomineralization on 6.5, 7.4 and 7.5 teeth.

**Figure 7 ijerph-18-08957-f007:**
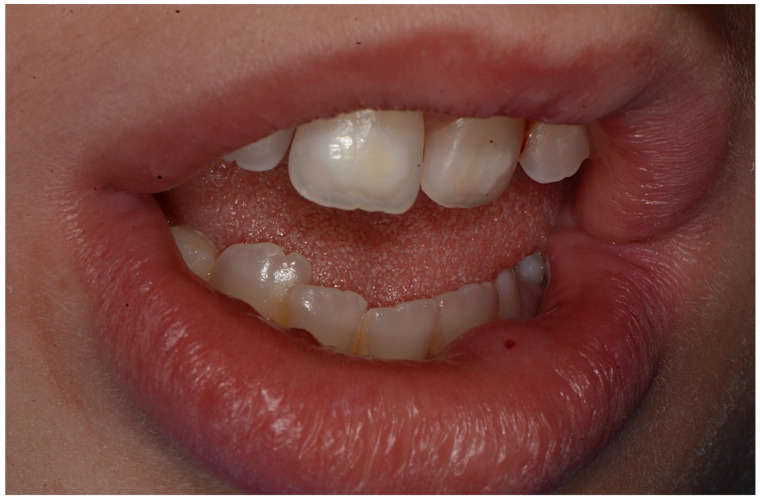
Incisor hypomineralization, dental malposition and presence of mucocele on lower lip.

**Table 1 ijerph-18-08957-t001:** Oral manifestation of HVDAS reported in Literature.

		Sex	Oral Manifestations	Age (Years)
Helsmoortel et al., 2014	Patient 1	M	thin upper lip	4.6
	Patient 2	F	thin upper lip, oligodontia	6.10
	Patient 3	F	N.D.	8
	Patient 4	M	small mouth with thin lips, small teeth and minor tongue tie	4.11
	Patients 5	M	normal palate, normal teeth	8.7
	Patient 6	M	largemouth widely spaced teeth	8.5
	Patient 7	F	thin upper lip	5
	Patient 8	M	high narrow palate	10.8
	Patient 9	F	N.D.	10.6
	Patient 10	M	N.D.	5.6
Pascolini et al., 2018	Patient n.2	F	small mouth, dental anomalies with multiple caries.	3.9
Van Dijck A et al., 2019	78 patients	M:F = 44:34		8.2 (mean)
Present study 2021	1 patient	M	high narrow palate, crowding, molar incisor hypomineralization (MIH), second class	9

M = Male, F = Female, N.D. = Not described.

## Data Availability

The data presented in this study are available on request from the corresponding author. The data are not publicly available due to privacy.

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
