# Peer review of "Oro-Dental Manifestations in a Pediatric Patient Affected by Helsmoortel-Van der Aa Syndrome"

_ijerph, 2021, doi:10.3390/ijerph18178957_

Round 1
Reviewer 1 Report
Author and Editor
Thank you for your work. I m generally happy with the content of this case report. but a number of points need clarifying before publish. These are given below.
Please provide the purpose of your study.
Please add full name of ADPN gene in line no 34 and mention as ADPN gene in all text as this terminology as mentioned previously.
Please clearly describe the symptoms, diagnosis, treatment and outcome.
M&M
Please include the methods to identify the gene in your study.
Conclusion
- Please add a clearly defined teaching objective, and elaborate whether this has been achieved. (Conclusions be drawn directly and accurately from observed results and not overgeneralized to populations and conditions not used in the study)
Figures
Fig 1 – please use arrow to show the ecchymosis, otherwise, use a clear picture
It’s better to use full name in all figures instead of using terminology
Table
What is the purpose of table 1? Please explain the clinical relevance of previously studies in the text.
Author Response
Dear Editor,
The Authors thank you and the reviewers for the opportunity to revise and improve this article. We genuinely believe that these corrections will raise the scientific value of this paper. To follow, you will find a point-to-point response to each suggestion of the referees. All the changes in the corrected manuscript are highlighted in blue.
We remain available for any further request.
dr Fedora della Vella
Reviewer 1
Thank you for your work. I m generally happy with the content of this case report. but a number of points need clarifying before publish. These are given below.
Reviewer: Please provide the purpose of your study.
Our response: The aim of the study was better expressed both in the abstact (“Aim of this case report is to describe oro-facial abnormalities in a patient…”), and at the end of the Introduction section (line 59: “We present a case of HVDAS patient highlighting the syndromic….”)
Reviewer: Please add full name of ADPN gene in line no 34 and mention as ADPN gene in all text as this terminology as mentioned previously.
Our response: The extended definition of ADNP gene( Activity-dependent neuroprotector homebox gene) is now specified at the first appearance of this term ( line 39 ) as suggested
Reviewer: Please clearly describe the symptoms, diagnosis, treatment and outcome.
Our response: A paragraph was added according to the reviewer suggestion (line 55)
Reviewer: Please include the methods to identify the gene in your study.
Our response: Fully detailed methods have now been added to the text.
Reviewer: Please add a clearly defined teaching objective, and elaborate whether this has been achieved. (Conclusions be drawn directly and accurately from observed results and not overgeneralized to populations and conditions not used in the study)
Our response: A conclusion section has been added addressing all the suggestion of the Reviewer
Reviewer: Fig 1 – please use arrow to show the ecchymosis, otherwise, use a clear picture
Our response: Corrected as suggested
Reviewer: It’s better to use full name in all figures instead of using terminology
Our response: Corrected as suggested
Reviewer: What is the purpose of table 1? Please explain the clinical relevance of previously studies in the text.
Our response: Table was commented at line 163 as suggested (“Most common oro-facial alterations…”)

Reviewer 2 Report
The present manuscript aimed to describe orofacial abnormalities in a patient affected by Helsmoortel- Van der Aa Syndrome. Despite the literature already present some previous reports regarding patients affected by Helsmoortel- Van der Aa Syndrome, the present case report can contribute with evidential information about the orofacial aspects.
Title:
Please describe that this is a pediatric patient;
Introduction:
Please insert the reference that have described the syndrome’s characteristics at the end of first paragraph.
“Few data have been reported on oro-dental manifestations in HVDAS patients.” Please be more specific and describe these previously reported data.
Case report:
Any research, involving human subjects, requires approval by a corresponding ethical compliance body. Please describe the ethical approval for the present case report.
Insert in the table 1, the characteristics observed in the present case report.
Discussion:
The authors believe that these observed orofacial abnormalities will be observed also in the secondary dentition? Please improve your discussion with this information.
Author Response
Dear Editor,
The Authors thank you and the reviewers for the opportunity to revise and improve this article. We genuinely believe that these corrections will raise the scientific value of this paper. To follow, you will find a point-to-point response to each suggestion of the referees. All the changes in the corrected manuscript are highlighted in blue.
We remain available for any further request.
dr Fedora della Vella
Reviewer 2
The present manuscript aimed to describe orofacial abnormalities in a patient affected by Helsmoortel- Van der Aa Syndrome. Despite the literature already present some previous reports regarding patients affected by Helsmoortel- Van der Aa Syndrome, the present case report can contribute with evidential information about the orofacial aspects.
Reviewer: Please describe that this is a pediatric patient.
Our response: The title was changed according to the Reviewer’s suggestion
Reviewer: Please insert the reference that have described the syndrome’s characteristics at the end of first paragraph.
Our response: Added as suggested
Reviewer: “Few data have been reported on oro-dental manifestations in HVDAS patients.” Please be more specific and describe these previously reported data.
Our response: We developed this aspect at line 163 as suggested, also addressing the table about previous studies’ data
Reviewer: Any research, involving human subjects, requires approval by a corresponding ethical compliance body. Please describe the ethical approval for the present case report.
Our response: Dear Reviewer, according to the Italian law case reports do not require the local Ethical Committee approval, since they do not involve real experimentations, but only describe observed clinical features. We already reported this aspect to the Editor during the submission process, enclosing the Declaration of Helsinki conformity statement and the disclosure about the patient’s privacy policy. If needed, we will provide this documentation to you too.
Reviewer: Insert in the table 1, the characteristics observed in the present case report.
Our response: Added as suggested
Reviewer: The authors believe that these observed orofacial abnormalities will be observed also in the secondary dentition? Please improve your discussion with this information.
Our response: At the time of our exam, the patients had mixed dentition and he presented with MIH on both primary and secondary teeth. We specified this aspect in the discussion as suggested (line 117 and 157).
Round 2
Reviewer 1 Report
Thank you for revised manuscript. My part is finished. However, please check the Title; it should be Oro-dental manifestations in a pediatric patient affected by .............. Syndrome. Right?
This manuscript is a resubmission of an earlier submission. The following is a list of the peer review reports and author responses from that submission.